# Comment on Hogas et al. Salt, Not Always a Cardiovascular Enemy? A Mini-Review and Modern Perspective. *Medicina* 2022, *58*, 1175

**DOI:** 10.3390/medicina59010051

**Published:** 2022-12-27

**Authors:** Norm R. C. Campbell, Francesco P. Cappuccio, Rachael M. McLean, Feng J. He, Graham A. MacGregor

**Affiliations:** 1Department of Medicine, University of Calgary, Calgary, AB T2N 2T9, Canada; 2WHO Collaborating Centre for Nutrition, Warwick Medical School, University of Warwick, Coventry CV4 7HL, UK; f.p.cappuccio@warwick.ac.uk; 3Department of Preventive & Social Medicine, University of Otago, Dunedin 9016, New Zealand; rachael.mclean@otago.ac.nz; 4Wolfson Institute of Population Health, Barts and The London School of Medicine & Dentistry, Queen Mary University of London, London EC1M 6BQ, UK; f.he@qmul.ac.uk (F.J.H.); g.macgregor@qmul.ac.uk (G.A.M.)

Hogas et al. recently published their perspective on dietary salt in a mini review [1]. The review outlined the broad support for reducing dietary salt from repeated and extensive governmental and nongovernmental scientific reviews. However, in their own review of evidence, Hogas et al. did not focus on the strong scientific basis for reducing dietary sodium [2]. That evidence has been recently reviewed in a ‘Global Call to Action’ supported by 75 international and national health and scientific organizations. In a meta-analysis of randomized controlled trials with post-trial follow-up, the reduction in sodium intake from 3646 mg/day to 2690 mg/day (1 g sodium = 2.5 g salt) was associated with a 26% reduction in cardiovascular disease (CVD) and a linear reduction in CVD was found between dietary sodium levels of 2300 to 4100 mg/day [2]. Further, a meta-analysis of cohort studies that assessed sodium intake by multiple non-consecutive 24 h urine samples reported a linear association between sodium intake (1846 to 5230 mg/day) and CVD [3]. Other meta-analyses of cohort studies that use quality criteria to exclude low-quality research methodology also reported a consistent finding of benefit from lowering dietary salt [2].

Much of the controversy relating to reducing dietary salt has been attributed to low-quality research prone to spurious findings and the propagation of misinformation on dietary sodium [2,4,5,6,7,8,9,10]. Many ‘controversial’ studies, such as the highly promoted Prospective Urban Rural Epidemiology (PURE) study, used a single-spot urine sample with a formula to estimate 24 h urine sodium levels [4,5]. The use of this method has been shown to cause spurious associations with blood pressure and mortality [9,11,12]. Single-spot urine samples with a formula to estimate 24 h urine sodium in individuals have large random and systematic errors, are not reproducible and have been strongly recommended not to be used by major scientific organizations [9,13]. Important methodological limitations have also been found in many other studies that report harms from lowering dietary salt [7]. Hogas et al. cite and quote a recent review by PURE study investigators that has been highly critiqued as providing inaccurate and misleading statements and improbable hypotheses based on the best available evidence [10]. Hogas et al. did not note the methodological limitations of the controversial data they present and also make two claims that the PURE investigators review has been endorsed by the European Society of Cardiology. We cannot find such an endorsement in that publication and the European Society of Cardiology has confirmed that it has not endorsed the review or even been asked to consider it for endorsement [10]. In contrast, the European Society of Cardiology Hypertension Council is a supporter of the Global Call to Action on dietary sodium [2] and the European Society of Cardiology recommendations are consistent with their support for reducing dietary sodium and are not consistent with the reviews by Hogas et al. [1] and the PURE investigators [10].

The review by Hogas et al. also cites a meta-analysis which reported an increase in all-cause mortality from salt restriction in patients with heart failure that included a trial from a center in Italy [14]. A meta-analysis of studies from the Italian center reported that two of the studies on heart failure contained duplicate data and when data verification was requested, it was indicated that the data had been lost in a computer failure leading the journal *Heart* to retract a meta-analysis of those studies (https://retractionwatch.com/2013/05/02/heart-pulls-sodium-meta-analysis-over-duplicated-and-now-missing-data/ accessed on 30 September 2022). Hogas et al. do not report a recent relatively large randomized controlled trial of sodium reduction in heart failure which reported an improvement in quality of life and NYHA class but no change in clinical events [15].

The World Health Organization states reducing dietary sodium is one of the most cost-effective interventions to improve population health [2]. Current high levels of dietary sodium were attributed to over 1.8 million deaths and over 44 million years of disability (DALYS) in the most recent report by the Global Burden of Disease Study [2]. It is of great public health importance to disseminate the best available evidence on dietary sodium.

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
