# Peer review of "Comment on Hogas et al. Salt, Not Always a Cardiovascular Enemy? A Mini-Review and Modern Perspective. Medicina 2022, 58, 1175"

_medicina, 2022, doi:10.3390/medicina59010051_

Round 1

Reviewer 1 Report

Dear authors,

Interesting comments regarding the paper Hogas et al. published previously. You've made the most critical points of this communication wonderfully well-structured. There are minor typos that need to be modified ( i.e. et al.).

Author Response

We greatly appreciate the reviewers comments.  Our spell check had not detected the 'et. al.' error .   This has been corrected as well as some spacing errors between sentences.

Reviewer 2 Report

Medicina Review:
Concerns about the Mini-Review and Modern Perspective on dietary salt
Headings/Page/Line - 1 (16-17)

Phrasing: "However, in their own review of evidence, Hogas et al have overlooked the strong scientific basis for reducing dietary sodium (2)."

Comments:

Suggest rephrase/soften the phrasing as Hogas did agree/mention about the established benefits scientific basis of low/reducing sodium BUT the objective of the paper was to determine the potential/possible adverse/negative perspective/the other side of the coins of salt reduction. Hogas showed in table attached below. Agree with comments (Campbell et al), that the evidence laid -‘Global Call to Action’ should be added by Hogas.

Author Response

We thank the reviewer for their comments.  We have softened the wording as suggested. It now reads 'Hogas et. al. do not focus on the strong scientific basis for reducing dietary sodium'.

Round 2

Reviewer 2 Report

Dear authors
Thank you for the updates from the first review. The authors have addressed the first comment. 

However, other following concerns addressed by the reviewer in the comments weren't answered. Appreciate authors consider the suggestion or points given. Or at plausible cause to justify/ reply by answering the comments. 

The attached file is the same comments file uploaded previously. Please ignore comment no.1 as it has been addressed. Thank you. 

Author Response

Response to the reviewers’ comments

Reviewer Comment 1 page 1 line 16-17

The reviewer implies that the mini review has an objective “to determine the potential/possible adverse/negative perspective/the other side of the coins of salt reduction”.  We find no statements in the Mini Review that support this being an overt objective.  Rather it overtly claims to be a “Mini-Review and modern perspective” and to provide “current literature regarding the advantages and disadvantages in the general population with knowledge gaps and up-to-date recommended interventions”.  As per our critique, it misrepresents mainstream science.  Our concern is not that our recent publication “Global Call to Action” was not cited, it is that none of the strong evidence supporting dietary salt reduction is cited.  The Global Call to Action concisely summarized this evidence.

The Reviewer suggests we soften our terms implying the review has citations on the evidence that reducing dietary sodium has benefits.  The reviewer graciously provided the exact citations in a screenshot of Table 1 from the paper.  This suggestion for revision is very puzzling as none of the citations mentioned in the mini review are those used in mainstream scientific publications to support sodium reduction but several of them are used by dissenting scientists to oppose sodium reduction. The citations indicated in table 1 are largely old, outdated and have been heavily critiqued.  None of the citations indicated address the strong science that favours sodium reduction. We have not revised our statement.

Reviewer comment 2 line 29-30

The reviewer indicates “The references (2, 4-6) given were all from the same author (Campbell et al), suggest to look for other evidences/literatures as self-citation may post to biases”.  We agree with the reviewer and have added references from other groups. 

Reviewer comment 3

Proposed author to rebut/comment based on the adverse events of low salt based on LR suggested below/ other potential LR to highlight that despite not always cardiovascular enemy BUT other potential issues such as high potassium/ hormone/lipids side effects/ unknown potential long term effects

We have not revised. Our letter is a critique of a misleading review.  We have highlighted a few of the many issues in the mini review.  If the editor wishes, they can invite original reviews on topics that include literature reviews to delve into the mainstream scientific details of interventions like low salt substitutes. In our view, our short critique is not a forum for detailed scientific review on specific interventions.  We note there are published reviews on the topics raised by the reviewer that reflect a more mainstream scientific perspective.  Specifically, more recent reviews find no RCT evidence of long-term stimulation of the RAS system in interventions lasting longer than 4 weeks and most of the global population has a lack of dietary potassium. We have not revised.

Reviewer comment 4 line 31-41.

Spot urine sodium versus 24 hr. urine sodium was not in the discussion in Hogas paper. The evidence on spot urine sodium has been validated/at par with 24 hr. urine sodium

The reviewer claims spot urine vs 24-hour urine was not in the discussion and yet the mini review had a focus on studies and reviews that used spot urine methodology. The reviewer makes a claim of validity of the spot sample that sharply conflicted with mainstream scientific interpretation. Systematic reviews of spot urine samples vs 24 hr. urine samples indicate spot samples are not a valid method for estimating individual sodium intake and a consortium of international health and scientific organizations specifically recommends against its use for this purpose (reference 13 Campbell NRC, He FJ, Tan M, Cappuccio FP, Neal B, Woodward M, et al. The International Consortium for Quality Research on Dietary Sodium/Salt (TRUE) position statement on the use of 24-hour, spot, and short duration (<24 hours) timed urine collections to assess dietary sodium intake. J Clin Hypertens. 2019;21:700-9).   Recently, the International Society of Hypertension, World Hypertension League and Resolve to Save Lives have formally endorsed the following position statement “It is strongly recommended to not conduct, fund, or publish research studies that use spot urine samples with estimating equations to assess individuals’ sodium (salt) intake in association with health outcomes”   The position statement was deemed necessary because of repeated false claims of validity such as those provided by the reviewer. The position is currently being supported other international organizations for publication in the New Year (WHL newsletter #178 Dec 2022). We have not revised.

Reviewer Comment 5 line 51-60

Author questioned the quality/rigorousness of Reference 12.  Cochrane review is an established database for M-A. It went through comprehensive and rigorous checklist/processes.  In additional the conclusion of this paper was appropriately cited by Hogas.

The reviewer ignores the issue we raised with the review - that it included a study where there is significant doubt about its integrity, with a meta-analysis that included the study being retracted due to duplicate data and a bizarre complete loss of 5 studies data in a computer crash making it impossible to validate the study data.  Further, this older meta-analysis was superseded by several subsequent meta-analyses including by the recent National Academies of Sciences, Engineering and Medicine as outlined in the Global Call to Action. We have not revised.

Reviewer Comment 6

Author commented on ref 13 on the improvement of QOL and NYHA class, BUT the objective of the paper was to look at clinical events. In which the outcome showed, low sodium did not reduce/ change clinical events

We respectfully disagree, that changes in quality of life and function are not important endpoints and also, we cannot find any evidence to support the reviewers claim that the authors defined that the objective was to only examine ‘hard’ clinical events.  Hogas et al specifically state that they aim to discuss effects of salt restriction “from a health point of view”. We agree with the reviewer, as per our critique, that Hogas et al should have examined some recent evidence on heart failure.  We have not revised.